# Evaluation of the Microbiological Effect of Colloidal Nanosilver Solution for Root Canal Treatment

**DOI:** 10.3390/jfb13040163

**Published:** 2022-09-25

**Authors:** Svetlana Razumova, Anzhela Brago, Haydar Barakat, Ammar Howijieh, Alexander Senyagin, Dimitriy Serebrov, Zoya Guryeva, Yuliya Kozlova, Elvira Adzhieva

**Affiliations:** 1Department of Propedeutics of Dental Diseases, Medical Institute, Peoples’ Friendship University of Russia (RUDN University), 6 Miklukho-Maklaya Street, 117198 Moscow, Russia; 2Department of Microbiology and Virology, Medical Institute, Peoples’ Friendship University of Russia (RUDN University), 6 Miklukho-Maklaya Street, 117198 Moscow, Russia

**Keywords:** antibacterial effect, bacteria, microorganisms, nanosilver

## Abstract

Background: The goal of endodontic treatment, along with the preparation of the root canal and giving it a shape corresponding to the obturation technique, is the drug treatment of the canal. The aim of this study was to determine the antibacterial effect of a colloidal solution of nanosilver at its various dilutions on root canal microorganism. Materials and methods: A solution of silver nanoparticles at a concentration of 10,000 ppm (1.0%) was diluted in various concentrations (10 solutions from 1% to 0.0025%). Cultures used for research: Str. agalacticae ATCC 3984, E. faecalis ATCC 323, St. aureus ATCC 4785, C. albicans ATCC 10231. After thawing, cultures of microorganisms were introduced into a liquid nutrient medium: cerebral heart broth for bacterial cultures and Sabouraud broth for C. albicans. The cultivation was carried out at a temperature of 37 °C for 24 h. A bacterial suspension for inoculation was prepared from a microbial sediment according to a turbidity standard of 0.5 McFarland in saline. Then, 100 μL of the obtained suspension of microorganisms was inoculated by the “lawn” method using a spatula on the Muller–Hinton medium. Solutions of silver nanoparticles were introduced into wells prepared in agar with a sterile metal punch. Further incubation was carried out for 24 h at 37 °C. Results: colloidal solution of silver nanoparticles at concentrations of 1%, 0.75%, 0.5% inhibited the growth of Str. agalacticae ATCC 3984 with a growth retardation zone of 6–7 mm. The E. faecalis ATCC 29212 strain was sensitive to solutions of silver nanoparticles at concentrations of 1%, 0.75%, 0.5% with a growth inhibition zone of 6–7 mm. Strain St. aureus 4785 demonstrated sensitivity to solutions of silver nanoparticles at concentrations of 1%, 0.75%, 0.5%, 0.1%, 0.05% with a growth retardation zone of 6-8 mm. Conclusion: colloidal solutions of silver nanoparticles have antimicrobial action against gram-positive bacteria (Str.agalacticae ATCC 3984, St. aureus ATCC 4785, E. faecalis ATCC 29212) and yeast-like fungi of the genus Candida (C. albicans ATCC 10231, C. albicans 672 and C. albicans D-225M), but this action is strain-specific and depends on the concentration of the solution.

## 1. Introduction

The goal of endodontic treatment, along with the preparation of the root canal and giving it a shape corresponding to the obturation technique, is the drug treatment of the canal [1]. Traditionally, a 1–3% sodium hypochlorite solution is used for drug treatment of the root canal, which has shown good clinical results. Long-term use of this solution in the clinic gives, according to various authors, success in endodontic treatment in 40–75% of cases [2]. However, in the treatment of necrotic forms of pulpitis and repeated endodontic treatment, many authors have suggested the use of antibacterial agents such as calcium hydroxide, chlorhexidine, MTAD [3]. Also, repeated endodontic treatment requires more careful adherence to the irrigation protocol with activation of hypochlorite not only by ultrasound, but also, for example, by heating or laser activation [4].

Long-term preservation of the results of antibacterial treatment of the walls of the root canal is the task of endodontic treatment, which today is solved by sealing restoration. 

The main idea in the preparation of the root canal is to achieve maximum disinfection of the dentinal walls of the root canal. Silver solutions were clinically proven agents, which showed an excellent antibacterial effect, but stained the dentin in a dark color, which limited their use in aesthetically significant areas [5,6].

Along with the development of nanotechnology, products with silver nanoparticles have appeared that can be used in endodontics. The appearance of colloidal solutions of nanosilver has opened up a new direction for root canal treatment.

Currently existing colloidal solutions of nanosilver differ in particle size or matrix. Therefore, many researchers have studied the biocompatibility of these solutions with body tissues [7,8,9].

The biocompatibility of a root canal irrigant based on positively charged imidazolium with an ionic liquid protected by nanosilver solution (AgNPs) was studied by Nabavizadeh et al., who concluded that the biocompatibility of NaOCl and chlorhexidine is lower compared to the nanosilver solution [8].

Generali et al. investigated the cytotoxicity and antimicrobial activity of irrigant solutions based on silver citrate BioAKT and BioAKT Endo. The cytotoxicity of various concentrations (0.25%, 0.5%, 1%, 2.5%, 5%) of these solutions (BioAKT and BioAKT Endo) was evaluated on L-929 mouse fibroblasts using the MTT assay. Both silver citrate solutions showed >70% mouse fibroblast viability when diluted to 0.25% and 0.5%. At higher concentrations, they were extremely cytotoxic. The FT-IR spectroscopy data of both liquids showed the same spectra, indicating similar chemical characteristics. Both solutions used as root canal irrigants showed significant antimicrobial activity and low cytocompatibility at dilutions greater than 0.5% [9].

Studies of the antibacterial activity of various forms of colloidal silver have been carried out by many researchers. Volkov et al., proposed endonanophoresis with a solution of hydrocolloidal silver Poviargol (Russia) and showed its clinical efficacy [7].

González-Luna et al., studied the treatment of a root canal with a nanosilver solution with a particle size of 10 nm. The results obtained showed that silver nanoparticles with a size of 10 nm and sodium hypochlorite at a concentration of 2.25% are effective for the elimination of E. faecalis, and there is no significant difference between them. The authors concluded that silver nanoparticles may represent a good option for removing E. faecalis from root canals [10].

Ioannidis et al. (2019) studied the antimicrobial efficacy of aqueous matrix synthesized graphene oxide (GO) (Ag–GO) silver nanoparticles (AgNPs) with various ex vivo irrigant delivery methods compared to conventional irrigants. As a result of the experiment, it was found that the effectiveness of the destruction of microorganisms with 2.5% NaOCl was higher compared to the experimental groups. The maximum destruction of the biofilm on the surface of the dentinal tubules was achieved by 2.5% NaOCl; however, Ag–GO caused a significant decrease in the total amount of biofilm volumes compared with the rest of the experimental groups [11].

According to the literature, the use of nanosilver hydrocolloid solutions from different manufacturers shows conflicting results on the antibacterial effect.

A new solution of colloidal nanosilver Argitos with particles of 1–2 nm, Russian-made NANOSPHERE company Russia (ARGITOS 10,000 ppm (1%), has appeared. To use this product in endodontic practice, it is necessary to study its antibacterial properties, so the aim of this study was to evaluate the antibacterial effect of a colloidal solution of Argitos nanosilver in its various dilutions.

## 2. Materials and Methods

To determine the effectiveness of using a colloidal solution with silver nanoparticles at a concentration of 10,000 ppm (1.0%) in the root canal, a microbiological research method was performed using the following cultures: Str. agalacticae ATCC 3984, Ent. faecalis ATCC 323, St. aureus ATCC 4785, C. albicans ATCC 10231. Cultures of microorganisms after thawing were introduced into a liquid nutrient medium: heart–brain broth (BHB, HIMEDIA^®^ M210, Mumbai, India) for bacterial cultures and Sabouraud broth (SDB, HIMEDIA® M033-500G, Mumbai, India) for C. albicans in a volume of 100 µL or 1 day (deposit for cryobox). The volume of BHB and SDB was 10 mL. Cultivation was carried out at a temperature of 37 °C for 24 h.

After cultivation, the nutrient medium together with the microorganisms were vortexed and 1 mL of the medium was taken from the test tube. This volume was added to an Eppendorf tube (V = 1.5 mL) and centrifuged for 10 min at 2.4 rpm (ELMI SkyLine CM-6M centrifuge, Sankt-Petersburg, Russia) to precipitate microorganisms from the broth. The supernatant was removed, and the culture pellet was resuspended with 0.9% sodium chloride solution. The bacterial inoculum suspension was prepared from the microbial pellet to a turbidity standard of 0.5 (McFarland, HIMEDIA^®^, Mumbai, India) in saline. Then, 100 μL of the suspension of microorganisms was inoculated by the “lawn” method using a spatula onto Petri dishes (dish diameter 90 mm) with Muller–Hinton medium (15 mL of medium).

To assess the antimicrobial activity of solutions with different concentrations of silver nanoparticles, the latter were introduced into wells made in agar with a sterile metal punch; Vwells = 18 µL, Dwells = 4.5 mm. Incubation was carried out for 24 h at 37 °C.

The mother liquor contained silver nanoparticles at a concentration of 10,000 ppm (1%). To prepare various dilutions of solutions, a colloidal solution of silver nanoparticles was diluted with distilled water. The following dilutions were prepared (Table 1):

Negative control: (K−), saline, (kd), distilled water. Positive control: (K+), CAC30/10 (ceftazidime 30 mcg/ clavulanic acid 10 mcg). Each experiment was repeated 5 times (n = 5). The results were evaluated by measuring the zones of growth inhibition around the well with the appropriate dilution of the silver nanoparticle solution.

After the results were obtained, yielding the discovery of a more pronounced antimicrobial activity against yeast-like fungi C. albicans, the study was expanded by adding two more clinical strains: C. albicans D-225M and C. albicans 672. All the experiments were carried out in quintuplets. The statistical significance was set at *p* ≤ 0.05. Student’s t test was carried out using the statistical software XLSTAT 2020 (Addinsof Inc., New York, NY, USA). The means and standard deviations were calculated, and the graph of inhibition zones was plotted by Microsoft Excel (Microsoft Excel for Office 365 MSO, Microsoft COP., Redmond, WA, USA).

## 3. Results

As a result of our research, it was found that a colloidal solution of silver nanoparticles at concentrations of 1%, 0.75%, 0.5% inhibits the growth of Str. agalacticae ATCC 3984 with a zone of growth inhibition − 6–7 mm. The strain E. faecalis ATCC 29212 was sensitive to solutions of silver nanoparticles at a concentration of 1%, 0.75%, 0.5% with a growth inhibition zone of 6–7 mm. Strain St. aureus 4785 showed sensitivity to silver solutions at a concentration of 1%, 0.75%, 0.5%, 0.1%, 0.05% with a growth inhibition zone of 6–8mm. The data are presented in Table 2.

As can be seen from Table 2, strains Str. agalacticae ATCC 3984 and E.faecalis ATCC 29212 were sensitive to a smaller range of silver nanoparticle solution concentrations (1%, 0.5%, 0.75%( compared to the St. aureus 4785 (1%, 0.75%, 0.5%, 0.1%, 0.05%).

The strain C. albicans ATCC 10231 demonstrated the highest sensitivity to solutions of silver nanoparticles. The zone of growth inhibition of the strain C. albicans ATCC 10231 around the well with solutions of silver nanoparticles ranged from 7 to 13 mm and had a direct dependence on the concentration of silver nanoparticles solution (0.5%, 0.75%, 0.1%, 0.05%, 0.025%, 0.01%, 0.075%): the higher the concentration of the solution of silver nanoparticles, the greater the zone of growth inhibition of the strain C. albicans ATCC 10231 around the well. The data are shown in Figure 1.

The clinical strain C. albicans 672 showed the highest sensitivity to solutions of silver nanoparticles. The zone of growth inhibition around the wells was observed in all concentrations of the solution of silver nanoparticles, and their range varied from 16 mm at a concentration of 0.05% to 10 mm at a concentration of 0.0025%, when taking into account the absence of a visible zone of inhibition around the positive control.

The clinical strain C. albicans D-225M showed the least sensitivity among yeast-like fungi. The effect of solutions of silver nanoparticles was observed only at concentrations of 1%, 0.75%, 0.5%, 0.1%. Zones of growth inhibition were in the range from 12 to 10 mm.

Common to C. albicans strains is that sensitivity was determined at concentrations of 1%, 0.5%, 0.75%, 0.1% with almost identical results.

## 4. Discussion

According to the results, solutions of silver nanoparticles have an inhibitory effect on the growth of all test cultures with a growth inhibition zone in the range from 6 to 16 mm. At the same time, in positive controls, the range of zones of growth inhibition is in the range from 13 to 24 mm.

Solutions of silver nanoparticles most actively inhibit the growth of yeast-like fungi of the genus Candida. The clinical strain C. albicans 672 was exposed to solutions of silver nanoparticles in the entire concentration range from 1.0% to 0.0025% with growth inhibition zones of 10–16 mm.

Farshad et al. (2016) studied the effect of a colloidal solution of nanosilver with imidazolium on the dentine surface. The authors found that the colloidal solution of silver affected the physicochemical properties of dentin and increased the roughness of its surface. The authors concluded that this irrigant can enhance the adhesion of bacteria and filling materials to the walls of the root canal dentin [12].

Rodrigues et al. evaluated the antimicrobial activity of an irrigant containing silver nanoparticles in an aqueous vehicle, sodium hypochlorite, and chlorhexidine against E. faecalis biofilm and infected dentinal tubules. The study was conducted on bovine teeth. The researchers concluded that the AgNP irrigant was not as effective against E. faecalis as compared to the NaOCl solution, which as an irrigant was more effective in disrupting the biofilm and killing bacteria in biofilms and dentinal tubules [13].

Tonini et al. presented the results of a study on the effectiveness of a new silver citrate-based root canal irrigation solution (BioAKT) in removing the smear layer, sealer penetration during canal filling, and the antibacterial activity of the new irrigant. BioACT and EDTA were the most effective solutions for both smear layer removal and sealer penetration. However, in the apical third, BioACT showed significantly better results compared to EDTA, both in removing the smear layer and sealing the canal. BioACT and sodium hypochlorite showed a comparable antibacterial effect. The investigators concluded that BioACT is a suitable smear layer remover that provides reliable sealer penetration into the apical part of the root canal system and has significant antibacterial properties [14].

The results of our study showed that Argitos colloidal nanosilver solution has an antimicrobial effect against C. albicans, Str. Agalacticae, E. faecalis, St. Aureus. The sensitivity of the E. Faecalis strain to colloidal silver solution is consistent with the data published by González-Luna, et al., Rodrigues et al. [10,13], and with data from Ioannidis et al., who found that the total volume of biofilms on a surface treated with silver is significantly lower compared to other antibacterial agents [11].

The data obtained allow us to conclude that solutions of silver nanoparticles have antimicrobial activity against gram-positive bacteria (Str. agalacticae ATCC 3984, St. aureus ATCC 4785, E. faecalis ATCC 29212), and yeast-like fungi of the genus Candida (C. albicans ATCC 10231, C. albicans 672, and C. albicans D-225M), but this effect is strain-specific and depends on the concentration of the solution. Further study of this drug in its effect on biofilms is needed. In the case of positive results, it will be possible to use a colloidal solution of silver as an irrigant for the final treatment of the root canal.

## 5. Conclusions

Within the limit of this study, colloidal solutions of silver nanoparticles have antimicrobial action against gram-positive bacteria (Str.agalacticae ATCC 3984, St. aureus ATCC 4785 E. faecalis ATCC 29212) and yeast-like fungi of the genus Candida (C. albicans ATCC 10231, C. albicans 672, and C. albicans D-225M), but this action is strain-specific and depends on the concentration of the solution.

## Figures and Tables

**Figure 1 jfb-13-00163-f001:**
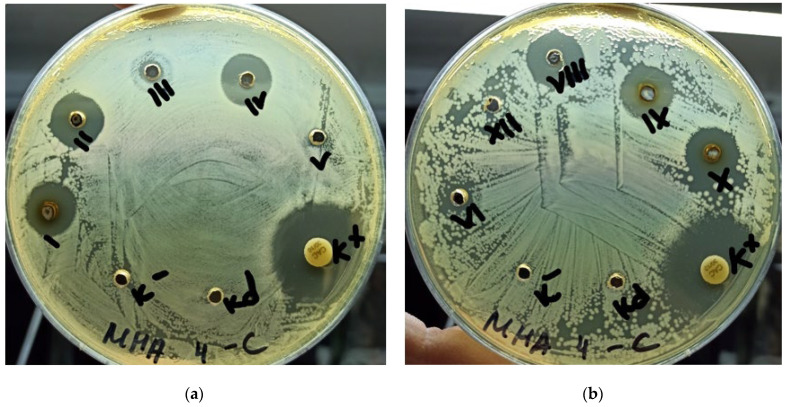
Growth of C.albicans ATCC 10231 on Muller–Hinton medium. (**a**) Numbers of solutions are signed in Roman numerals: I, % solution; II, 0.1% solution; III, 0.01% solution; IV, 0.05% solution; V, 0.005% solution; (**b**) VI, 0.025% solution; VII, 0.0025% solution; VIII, 0.075% solution; IX, 0.75% solution; X, 0.5% solution.

**Table 1 jfb-13-00163-t001:** List of solutions of colloidal nanosilver for research.

Solution Number	Final Ag Concentration (%)	Volumes of Solutions to Obtain the Final Concentration
No 1	1	100 µL of starting material
No 2	0.1	90 µL distilled water + 10 µL solution No 1
No 3	0.01	90 µL distilled water + 10 µL solution No 2
No 4	0.05	50 µL distilled water + 50 µL solution No 2
No 5	0.005	50 µL distilled water + 50 µL solution No 3
No 6	0.025	50 µL distilled water + 50 µL solution No 4
No 7	0.0025	50 µL distilled water + 50 µL solution No 5
No 8	0.075	25 µL distilled water + 75 µL solution No 2
No 9	0.75	25 µL distilled water + 75 µL solution No 1
No 10	0.5	50 µL distilled water + 50 µL solution No 1

**Table 2 jfb-13-00163-t002:** Sensitivity of microorganisms to various concentrations of nanosilver diluted with distilled water.

Strain	Zone of Inhibition (mm) at the Final Concentration of Nanosilver (%)										K−	K+	kd
	1	0.75	0.5	0.1	0.075	0.05	0.01	0.025	0.005	0.0025			
Str. agalacticae 3984	6.8 ± 0.4	6 ± 0	6.6 ± 0.5	0	0	0	0	0	0	0	0	15 ± 0	0
E.faecalis ATCC 29212	7 ± 0	5.8 ± 0.4	6.8 ± 0.4	0	0	0	0	0	0	0	0	13 ± 0	0
St. aureus 4785	8 ± 0	7.8 ± 0.4	7.6 ± 5.4	6.8 ± 0.4	6 ± 0	6 ± 0	0	0	0	0	0	18 ± 0	0
C. albicans ATCC 10231	12 ± 0	12 ± 0	12 ± 0	12.2 ± 0.4	12.4 ± 0.5	11.6 ± 0.8	7 ± 0	6.8 ± 0.4	0	0	0	24 ± 0	0
C. albicans 672	12 ± 0	12 ± 0	11.8 ± 0.4	14.4 ± 0.8	14.8 ± 0.4	15.6 ± 0.5	14.8 ± 0.4	12 ± 0	13.8 ± 0.4	9.8 ± 0.4	0	0	0
C. albicans D–225M	12 ± 0	12 ± 0	11.8 ± 0.4	10.2 ± 0.4	0	0	0	0	0	0	0	18 ± 0	0

## Data Availability

Not applicable.

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
