# Peer review of "Evaluation of the Microbiological Effect of Colloidal Nanosilver Solution for Root Canal Treatment"

_jfb, 2022, doi:10.3390/jfb13040163_

Round 1

Reviewer 1 Report

The manuscript “Evaluation the microbiological effect of colloidal nanosilver solution for root canal treatment” by Svetlana Raxumova et al. aims at understanding the role of microbiological effect of colloidal nanosilver in dental applications. The authors addressed a highly important question in the endodontic treatment by determining the antibacterial effect of a colloidal solution of nanosilver on root canal microorganisms To support their data, the authors used a combination of different tooth cavity-relevant microbe cultures and used the "lawn" method using a spatula on the Muller-Hinton medium. The authors find that the colloidal solutions of silver nanoparticles have concentration-dependent antimicrobial action against gram-positive bacteria and yeast-like fungi of the genus Candida.

This is a very interesting and timely paper and I have thoroughly enjoyed reading and reviewing this manuscript. The topic is of high importance and the effect of nanosilver has gained tremendous attention in recent years. Work data presented in the paper will increase our knowledge about this important aspect of applications of nanosilver. The experiments are well-designed, and the results are clean. The approach is simple and elegant, easy to follow, and the interpretation is convincing. The paper is also clearly written and accessible to a broad audience, and the figures are clear and helpful. I am rather enthusiastic about this paper and supportive of its publication. I only offer some minor suggestions to improve readability and enhance the message of the paper (adopting them is optional).

In sum, I very much enjoyed this very interesting paper, and I am positive about its publication.

Minor issues: 

  1. Authors have not mentioned the number of repeats for their experiments.  

  2. Authors should mention n, mean and standard deviation in the material section as well as in the table. 

  3. Authors should use appropriate statistical tests and mention that in the material section. 

  4. Authors should check the growth curve of different microbes at different nano-silver concentrations. 

Author Response

Dear Reviewer:

Authors have not mentioned the number of repeats for their experiments.  

Answer: We mentioned in the article

Authors should mention n, mean and standard deviation in the material section as well as in the table. 

Answer: We mentioned in the article

Authors should use appropriate statistical tests and mention that in the material section. 

Answer: We mentioned in the article

Authors should check the growth curve of different microbes at different nano-silver concentrations. 

Answer: (we have not performed this recommendation, we need more time about 2 weeks for this if you can give us this time we can perform it)

Reviewer 2 Report

Dear Authors, the manuscript is well written and organized but it needs some adjustments:
Abstract: line 36
"gram-negative bacteria (E. faecalis ATCC 29212)"
this must be correct: ""E. Faecalis is a gram-positive""

Introduction:
In the first part of the introduction, you need to explain more about "irrigant activation", which is a fundamental step in modern endodontics. This phase is crucial to decrease the bacterial load. About this, I suggest you add this reference:

Iandolo A, Abdellatif D, Pantaleo G, Sammartino P, Amato A. Conservative shaping combined with three-dimensional cleaning can be a powerful tool: Case series. J Conserv Dent. 2020 Nov-Dec;23(6):648-652. doi: 10.4103/JCD.JCD_601_20. Epub 2021 Feb 11. PMID: 34083925; PMCID: PMC8095685.

This article explains well the importance of irrigant activation.

Discussion:
In the discussion, I suggest explaining the importance of irrigant activation.

Line 210: "gram-negative bacteria (E. faecalis ATCC 29212)"

this must be correct: ""E. Faecalis is a gram-positive""

Conclusions:
Line 219:  "gram-negative bacteria (E. faecalis ATCC 29212)"

this must be correct: ""E. Faecalis is a gram-positive""

Author Response

Dear reviewer:

Dear Authors, the manuscript is well written and organized but it needs some adjustments:
Abstract: line 36
"gram-negative bacteria (E. faecalis ATCC 29212)"
this must be correct: ""E. Faecalis is a gram-positive"" (Corrected)

Introduction:
In the first part of the introduction, you need to explain more about "irrigant activation", which is a fundamental step in modern endodontics. This phase is crucial to decrease the bacterial load. About this, I suggest you add this reference:

Iandolo A, Abdellatif D, Pantaleo G, Sammartino P, Amato A. Conservative shaping combined with three-dimensional cleaning can be a powerful tool: Case series. J Conserv Dent. 2020 Nov-Dec;23(6):648-652. doi: 10.4103/JCD.JCD_601_20. Epub 2021 Feb 11. PMID: 34083925; PMCID: PMC8095685.

This article explains well the importance of irrigant activation.
(We added the reference)
Discussion:
In the discussion, I suggest explaining the importance of irrigant activation.

Line 210: "gram-negative bacteria (E. faecalis ATCC 29212)"

this must be correct: ""E. Faecalis is a gram-positive"" (Corrected)

Conclusions:
Line 219:  "gram-negative bacteria (E. faecalis ATCC 29212)"

this must be correct: ""E. Faecalis is a gram-positive"" (Corrected)

Reviewer 3 Report

Reviewer’s comments

Ms.ID jfb-1888774

Title:  Evaluation the microbiological effect of colloidal nanosilver solution for root canal treatment

Authors: Svetlana Raxumova, Anzhela Brago, Haydar Barakat, Ammar Howijieh, Alexander Senyagin, Dimitriy Serebrov, Zoya Guryeva, Yuliya Kozlova and Elvira Adzhieva

General Comments:

  The aim of this study is to determine the antibacterial effect of a colloidal solution of nanosilver at its various dilutions on root canal microorganism.  

  Before the acceptance to Journal of Functional Biomaterials, authors should add the followings;

1. In the Materials and Methods section and Table 2, authors compared a colloidal solution of nanosilver with ceftazidime/ clavulanic acid. Additionally, authors should compare it with the existing reagents of root canal (e.g., NaOCl, CHX). 

2. In the Introduction section and the Discussion section, the authors describe the effect of “biofilm”. However, the authors do not evaluate that. The authors should evaluate minimum biofilm inhibitory concentration (MBIC).

Author Response

General Comments:

  The aim of this study is to determine the antibacterial effect of a colloidal solution of nanosilver at its various dilutions on root canal microorganism.  

  Before the acceptance to Journal of Functional Biomaterials, authors should add the followings;

  1. In the Materials and Methods section and Table 2, authors compared a colloidal solution of nanosilver with ceftazidime/ clavulanic acid. Additionally, authors should compare it with the existing reagents of root canal (e.g., NaOCl, CHX). (the protocol for the study of the sensitivity of microorganisms to an antiseptic suggests a positive control, that is why we used this antibiotic)

  1. In the Introduction section and the Discussion section, the authors describe the effect of “biofilm”. However, the authors do not evaluate that. The authors should evaluate minimum biofilm inhibitory concentration (MBIC). (this was not included in the aim of our study)

Round 2

Reviewer 3 Report

I recommend the accept of this revised manuscript.